

# Intensified modulation of aerosol pollution in China by El Niño with short duration

Liangying Zeng[1], Yang Yang[1*], Hailong Wang[2], Jing Wang[3], Jing Li[4], Lili Ren[1],

Huimin Li[1], Yang Zhou[1], Pinya Wang[1], Hong Liao[1]

[1]Jiangsu Key Laboratory of Atmospheric Environment Monitoring and Pollution

Control, Jiangsu Collaborative Innovation Center of Atmospheric Environment and

Equipment Technology, School of Environmental Science and Engineering, Nanjing

University of Information Science and Technology, Nanjing, Jiangsu, China

[2]Atmospheric Sciences and Global Change Division, Pacific Northwest National

Laboratory, Richland, Washington, USA

[3]Tianjin Key Laboratory for Oceanic Meteorology, Tianjin Institute of Meteorological

Science, Tianjin, China

[4]Department of Atmospheric and Oceanic Sciences, School of Physics, Peking

University, Beijing, China

*Correspondence to yang.yang@nuist.edu.cn



## Abstract

El Niño-Southern Oscillation (ENSO), a phenomenon of periodic changes in sea surface temperature in the equatorial central eastern Pacific Ocean, is the strongest signal of interannual variability in the climate system with a quasi-period of 2-7 years. El Niño events have been shown to have important influences on meteorological conditions in China. In this study, the impacts of El Niño with different durations on aerosol concentrations and haze days during December-January-February (DJF) in China are quantitatively examined using the state-of-the-science Energy Exascale Earth System Model version 1 (E3SMv1). We find that $PM_{2.5}$ concentrations are increased by 1-2 µg m$^{-3}$ in the northeastern and southern China and decreased by up to 2.4 µg m$^{-3}$ in central-eastern China during El Niño events relative to the climatological means. Compared to long duration (LD) El Niño events, El Niño with short duration (SD) but strong intensity causes northerly wind anomalies over central-eastern China, which is favorable for aerosol dispersion over this region. Moreover, the anomalous southeasterly winds weaken the wintertime prevailing northwesterly in northeastern China and facilitate aerosol transport from South and Southeast Asia, enhancing aerosol increase in northeastern China during SD El Niño events relative to LD El Niño events. In addition, the modulation on haze days by SD El Niño events is 2-3 times more than that by LD El Niño events in China. The aerosol variations during El Niño events are mainly controlled by anomalous aerosol accumulation/dispersion and transport due to changes in atmospheric circulation, while El Niño-induced precipitation change has little effect. The occurrence frequency of SD El Niño events has been increasing significantly in recent decades, especially after 1940s, suggesting that El Niño with short duration have exerted increasingly intense modulation on aerosol pollution in China over the past few decades.



## 1. Introduction

Since the beginning of the 21st century, China has experienced frequent events of heavy haze pollution (Yang et al., 2018). The excessive aerosol concentrations during the heavy haze events can cause a large decrease in atmospheric visibility (Han et al., 2013) and pose significant public health hazards, such as a dramatic increase in cardiovascular and respiratory diseases and associated mortality rates (Liu et al., 2019). $PM_{2.5}$ (particulate matter less than 2.5 μm in diameter) has been reported to be the fifth leading risk factor for mortality, inducing 7.6% of total deaths globally in 2015 (Cohen et al., 2017). In order to alleviate air pollution, China has been continuously taking clean air actions in the recent years to battle for the blue sky. If all goes as planned, by 2035, the quality of atmospheric environment will be fundamentally improved and the goal of a beautiful China will be basically achieved. However, this goal requires a comprehensive and better scientific understanding of all factors that can affect aerosol concentrations and haze pollution in China.

Undoubtedly, the rise of anthropogenic emissions is the fundamental reason for the increase in aerosol concentration and haze pollution events (Huang et al., 2014), but the unfavorable meteorological condition, as one of the most important external factors, has been reported to have substantial influences on haze formation (Yang et al., 2016a; Wang et al., 2019, 2020a). With increasing greenhouse gases in the future (2050-2099), severe winter haze events in Beijing would become 50% more frequent and 80% longer in duration, compared to the historical period (1950-1999), due to an accelerated warming of the lower atmosphere and weakening of the East Asian winter monsoon (Cai et al. ,2017). In addition, external forcings, such as Pacific Decadal Oscillation (Zhao et al., 2016) and Arctic sea ice (Wang et al., 2015; Zou et al., 2020), all have important impacts on aerosol concentrations and haze pollution in China. El Niño-Southern Oscillation (ENSO), as another prominent climate phenomenon caused by the coupled atmosphere-ocean interactions in the tropical Pacific Ocean (Trenberth, 2019), is a significant signal of interannual climate change on a global scale. It triggers atmospheric circulation and precipitation anomalies globally (Yang et al., 2016b, 2016c) and certainly have an important impact on haze events and aerosol concentrations in China by modulating the East Asian winter monsoon system (Sakai and Kawamura, 2009; Wang et al., 2000; Zhang et al., 2017).

The ENSO cycle is composed of warm-phase (i.e., El Niño) and cool-phase (i.e.,



La Niña) of sea surface temperatures (SSTs) over the tropical eastern Pacific Ocean, which further cause precipitation, atmospheric circulation and temperature anomalies in much of the tropics and subtropics. Such changes also affect the spatiotemporal distribution of aerosols in China (Feng et al., 2017, 2020; Sun et al., 2018; Yang et al., 2014; Zhao et al., 2018; Zhu et al., 2012; Wang et al., 2020b). During a strong El Niño event in 2015/2016, $PM_{2.5}$ concentrations in winter were observed to increase by 20-100 $\mu g/m^3$ in eastern China compared to that in 2014, which was attributed to the weakened wind speed in the North China Plain during the El Niño event (Chang et al., 2016; Wang et al., 2020a). $PM_{2.5}$ concentrations in southern China were also decreased by about 20 $\mu g/m^3$ during the 2015/2016 El Niño event, which was attributed to an enhanced precipitation and aerosol wet scavenging over this region. Based on haze day counting mainly using atmospheric visibility, many studies found that El Niño events could induce more (fewer) winter haze days in northern (southern) China (Gao and Li, 2015; Li et al., 2017). In addition to surface observations, several studies have also analyzed the relationship between ENSO events and aerosol loading based on aerosol optical depth (AOD) data from satellite retrievals. Jeoung et al. (2014) analyzed the combined AOD data of MODIS (Moderate Resolution Imaging Spectroradiometer), MISR (Multi-angle Imaging SpectroRadiometer) and AERONET (Aerosol Robotic Network) and found that during the warm phase of ENSO, the fine-mode AOD increased in eastern coastal areas but decreased in some inland areas of China. Sun et al. (2018) studied the influence of ENSO events on the interannual variation of wintertime aerosol in China using AOD data (1980-2016) from MERRA-2 reanalysis and found that AOD in the North China Plain increased significantly during El Niño events, with a 15% increment in the Beijing-Tianjin-Hebei region compared to the long-term average. They also pointed out that AOD increased in eastern and southern China and decreased in southwestern China during El Niño events.

Although observational data showed that aerosols in China were largely perturbed during El Niño events, the individual impacts of atmospheric circulation and precipitation anomalies associated with El Niño could not be simply extracted out with observations alone. Numerical simulations have been used to isolate the individual impacts of El Niño on aerosols in China through a superposed SST perturbation method and explore the underlying mechanisms (Yu et al., 2019; Zhao et al., 2018). Based on an aerosol-climate coupling model, Zhao et al. (2018) suggested



that El Niño increased the seasonal mean aerosol concentration in southern China in
winter, which is mainly due to the increased aerosol transport from South and
Southeast Asia. Using the same model, Yu et al. (2019) showed that, relative to the
climatological mean, wintertime surface aerosol concentrations in northeastern and
southeastern China (central and southwestern China) increased (decreased) during El
Niño events, which was mainly attributed to anomalies in near-surface winds and the
resulting aerosol mass flux divergences. Sun et al. (2018) used the aerosol-climate
model CAM5 to simulate the impact of ENSO events on the interannual variability of
AOD in China and found that El Niño events led to an increase in AOD in central and
eastern China. They suggested that the change in AOD was mainly dominated by the
change in meridional winds.
Some studies focused on the effects of different spatial types (e.g., East Pacific
and Central Pacific El Niño, Kao and Yu (2009)) and intensities of El Niño events on
aerosol concentrations in China (e.g., Yu et al., 2019). Yu et al. (2019) found that, due
to the difference in atmospheric circulation between two types of El Niño, Central
Pacific El Niño events resulted in a larger increase in aerosol burden in southern
China than East Pacific El Niño events. They also indicated that a moderate El Niño
event led to an increase in seasonal mean near-surface aerosol concentrations
throughout eastern China in winter, while a strong or weak El Niño event brought
about a significant decrease in aerosol concentrations in northern China.
Apart from the spatial types, El Niño can also be categorized as short duration
(SD) and long duration (LD) according to the length of their decay period (Boo et al.,
2004; Chen et al., 2012; Guo and Tan, 2018). These two temporal types of El Niño
events have been confirmed to have different impacts on the SSTs, vertical wind shear,
relative humidity and precipitation in South China Sea and Philippine Sea (Guo and
Tan, 2018; Wu et al., 2019). The El Niño events with different durations are likely to
have different impacts on the aerosol distribution in China. However, few studies
explore the different impacts of SD and LD El Niño events on aerosol concentrations
and haze days in China, as well as the associated mechanisms, which are essential for
air pollution control in the near future.
In this study, the effects of SD and LD El Niño events on wintertime aerosols in
China are investigated by using the state-of-the-science Energy Exascale Earth
System Model (E3SM). The data, model, and analysis methods used in this research
are presented in Section 2. The influences of different durations of El Niño events on



aerosols over China and the mechanisms involved are analyzed in Section 3.
Summary of the main results and discussion of the implications for future research are
provided in Section 4.

## 2. Data and Methods

### 2.1 Data

We use the following datasets in this study.
(1) The merged Hadley-NOAA/OI SST and sea ice concentration (SIC) datasets
from 1870 to 2017 with a horizontal resolution of $1° \times 1°$ (Hurrell et al., 2008)
are used to obtain the climatological mean SST and SIC pattern and the
anomalies of SST during SD and LD El Niño events.
(2) Monthly mean emissions of aerosols and their precursors in 2014 from the
CMIP6 (Coupled Model Intercomparison Project Phase 6) (Hoesly et al.,
2018; van Marle et al., 2017) with emissions in China replaced by MEIC
(multi-resolution emission inventory for China) emission inventory are used
as input datasets in model simulations.
(3) Hourly observations of $PM_{2.5}$ concentrations at 1657 stations over China
from December 2014 to February 2015 derived from the China National
Environmental Monitoring Centre (CNEMC) (http://www.cnemc.cn) are
applied to evaluate the model performance.

### 2.2 SD and LD El Niño events

Here we first describe how the LD and SD El Niño events are defined. The year
in which El Niño developed is denoted by $year^0$ and the months of that year is denoted
by $Jan^0$, $Feb^0$, ..., and $Dec^0$, while the following year and months are $year^1$ and $Jan^1$,
$Feb^1$, ...., and $Dec^1$, respectively. Each El Niño event is firstly identified when a
3-month running mean Niño 3.4 index, defined as the regional mean linear detrended
SST anomaly from the monthly mean climatology over the Niño3.4 region
(170°W-120°W, 5°S-5°N), is greater than 0.75°C in any month from $Oct^0$ to $Feb^1$ of
the developing phase. If the Niño 3.4 index is higher than 0.5°C in any month from
$Oct^1$ to $Feb^2$ of the decaying phase, the El Niño event is an LD El Niño event;
otherwise, it is an SD El Niño event (Wu et al., 2019).
Figure 1 shows the time series of the Niño3.4 indices calculated based on the
Hadley-NOAA/OI data for the period 1870-2017. According to the definition above,
totally 30 El Niño events are identified in this time period, with 22 SD El Niño events





182    (1877/1878, 1885/1886, 1888/1889, 1896/1897, 1902/1903, 1911/1912, 1923/1924,

183    1925/1926, 1930/1931, 1951/1952, 1957/1958, 1963/1964, 1965/1966, 1972/1973,

184    1982/1983, 1991/1992, 1994/1995, 1997/1998, 2002/2003, 2006/2007, 2009/2010,

2015/2016 as developing phase) and 8 LD El Niño events (1899/1900, 1904/1905,
1913/1914, 1918/1919, 1939/1940, 1968/1969, 1976/1977, 1986/1987 as developing
phase). The temporal evolution of the Niño3.4 indices during the developing and
decaying phases of SD and LD El Niño events are shown in Figure 2. During the
developing phase from $Jul^0$ to $Feb^1$, due to the fast accumulation of ocean heat content
and rapid adjustments of the surrounding seas to the tropical Pacific Ocean warming
(Wu et al., 2019), the Niño3.4 indices are higher in SD El Niño events, but the SST
decreases rapidly in the decaying phase, compared to those in LD events.

### 2.3 Model description and experimental design

To quantify the influence of El Niño with various durations on aerosols in China,

the U.S. Department of Energy (DOE) E3SM version 1 (E3SMv1) is utilized in this
study, which includes atmosphere, land, ocean, sea ice and river components (Golaz et
al., 2019). It was branched from the CESM1 (Community Earth System Model) but
has been updated substantially since. The E3SM Atmosphere Model version 1
(EAMv1) is a descendant of the well-known CAM5.3 (Community Atmosphere
Model version 5.3) (Rasch et al., 2019). It includes considerable upgrades to aerosols,
turbulence, chemistry and cloud related processes. EAMv1 provides various options
of spatial resolution. In this study, the horizontal spatial resolution of approximately
1° and 30 vertical layers from the surface to 3.6 hPa are used in the model
configuration. The model simulates aerosols including sulfate, black carbon (BC),
primary organic aerosol (POA), secondary organic aerosols (SOA), sea salt, and
mineral dust in the four-mode Modal Aerosol Module (MAM4) (Wang et al., 2020).

The following numerical experiments are conducted in this study. A "CLIM"

experiment is driven by climatological average of monthly SST and SIC over
1870-2017 and integrated for 20 years. Two sets of sensitivity experiments, "SD" and
"LD", are respectively driven by the monthly SST representing composite SD and LD
El Niño events. The monthly SSTs representing SD (LD) El Niño events are produced
through superposing the average of monthly SSTs from $Jul^0$ to $Jun^1$ of the 22 SD (8
LD) El Niño events selected in Sec. 2.2 on top of the climatological monthly SST
over 60°S-60°N. Each set of sensitivity experiment has 3 ensemble members with
different initial conditions branched from different years of the CLIM experiment.





Each member of the sensitivity experiments is run for 8 years with first 3 years used
for spin-up and the last 5 years used for analysis. The differences in the monthly and
daily mean model fields between SD, LD and CLIM are used to analyze the effects of
duration of El Niño events on aerosols. To understand the potential mechanism of El
Niño impacts on aerosol pollution in China, two additional experiments are also
conducted. The "SD_emis" experiment is the same as the first ensemble member of
SD experiment, except that the emissions of aerosols and precursor gases from South
and Southeast Asia are turned off. The "CLIM_emis" experiment is same as the
"SD_emis" experiment but driven by climatological average of monthly SST and SIC
over 1870-2017. All other external forcings, including insolation, greenhouse gas
concentrations, and emissions of aerosol and precursor are kept at present-day
conditions (year 2014), In brief, the experiments performed are as follows.
1. CLIM: control simulation driven by climatological SST.
2. SD: sensitivity simulation to quantify the impacts of El Niño events with
short duration on aerosols in China. Same as CLIM except for the imposed
SST pattern of short duration El Niño.
3. LD: sensitivity simulation to quantify the impacts of El Niño events with
long duration on aerosols in China. Same as CLIM except for the imposed
SST pattern of long duration El Niño.
4. SD_emis: sensitivity simulation to quantify the role of regional transport of
aerosols from South and Southeast Asia on aerosols in China during El Niño
events with short duration. Same as SD except that the emissions of aerosols
and precursor gases from South and Southeast Asia are turned off.
5. CLIM_emis: sensitivity simulation to serve as the baseline for SD_emis.
Same as SD_emis except for the use of climatological SST.

## 2.4 Model evaluation

To evaluate the model performance in simulating aerosol concentration and
distribution in China, the simulated December-January-February (DJF) mean surface
PM$_{2.5}$ (sum of sulfate, BC, POA and SOA in model simulation) concentrations from
the CLIM experiment is compared with the observed PM$_{2.5}$ concentrations in Fig. 4.
The model well reproduced the spatial distribution of wintertime aerosols in China,
with high aerosol concentrations in eastern China (e.g., North China Plain, Fenwei
Plain and Yangtze River Delta) and southwestern China (e.g., Sichuan Basin) and low
aerosol levels in western China. The spatial correlation coefficient (R) between the



E3SMv1 EAMv1 simulation and observations for near-surface PM$_{2.5}$ concentrations is
+0.43. However, the model underestimates the PM$_{2.5}$ concentrations in China, with a
normalized mean deviation (NMB) of -65.74% compared to the observed values,
which was also reported in many studies using the CESM1 model (e.g., Yang et al.,
2017a, b). The discrepancy could be due to many factors, including the lack of nitrate
and ammonium aerosols in the model, strong wet scavenging simulated at the mid-
and high latitudes, and less transformation from gas to particles. In addition, we focus
on anthropogenic aerosols. If natural dust is considered in the modeled PM$_{2.5}$
calculation, the NMB will drop to -6.38%. Nevertheless, the aerosol concentrations in
EAMv1 simulations are closer to the observations than previous ENSO-aerosol
studies (e.g., Yu et al., 2019; Zhao et al., 2018) and the composite differences are
analyzed in this study rather than the climatological mean concentration. We don't
expect the systematic low biases in PM$_{2.5}$ concentrations affect our study on the
impact of El Niño events. However, we should note that the aerosol changes in China
during SD/LD El Niño events in the real world could be larger than the simulated
values here.

## 3. Results

### 3.1 Impacts of SD and LD El Niño events on aerosol concentrations

Figures 5 and 6 show the absolute and relative impacts, respectively, of the two
types of El Niño events with different durations on the simulated DJF mean
near-surface concentrations and column burdens of PM$_{2.5}$ in China. The effects of the
SD and LD El Niño events on near-surface aerosol concentrations over China are
similar in the spatial pattern distribution, with increases in the northeastern and
southern China by about 1-2 µg m$^{-3}$ (5-15% compared to the climatological mean)
and decreases in central-eastern China during El Niño events relative to the
climatological averages. This spatial pattern of aerosol changes is in accordance with
previous modeling studies (Feng et al., 2016; Yu et al., 2019; Zhao et al., 2018).
However, the modeling results are not exactly the same as the observed PM$_{2.5}$ changes,
which show increases in PM$_{2.5}$ over northeastern China and the North China Plain and
slightly decreases in southern China during the 2015/2016 El Niño event (Chang et al.,
2016). The discrepancy between the model simulations and observations can be
attributed to the following reasons. First of all, instead of the El Niño impacts,
observed aerosols can be affected by other factors including East Asian winter
monsoon (Yang et al., 2016a), Arctic Oscillation (Zhang et al., 2019) and Pacific
Decadal Oscillation (Zhao et al., 2017), whereas the modeled changes are purely
caused by the El Niño impacts through the imposed SST perturbation. Secondly, the
time coverage of near-surface $PM_{2.5}$ observations is limited in China and only one
extreme El Niño event (2015/2016) was analyzed in previous El Niño-$PM_{2.5}$ studies
(e.g., Chang et al., 2016), which is not fully representative of the impact of general El
Niño events.
Although the spatial patterns of the SD and LD El Niño influences on the DJF
$PM_{2.5}$ concentrations in China resemble each other, the magnitudes of the influences
are different. Central-eastern China experiences more reductions in near-surface $PM_{2.5}$
concentrations during SD El Niño, with the concentration decreases of more than 2.4
$\mu g\ m^{-3}$ (15% relative to the climatological mean), which is much larger than the 0.6
$\mu g\ m^{-3}$ (5%) during LD El Niño. In southern China, the spatial coverage of the
increase in $PM_{2.5}$ concentration shrinks more during SD than LD El Niño events
relative to the CLIM, but the intensities of the anomalies triggered by the two
temporal types of El Niño events are similar. Moreover, SD El Niño induces a larger
increase in $PM_{2.5}$ concentrations in northeastern China by 1.2 $\mu g\ m^{-3}$ (10%) than that
of 0.6 $\mu g\ m^{-3}$ (5%) during LD El Niño events.
The $PM_{2.5}$ burden and near-surface concentration anomalies triggered by the El
Niño events with short and long durations are basically the same in spatial distribution
but with different magnitudes (Fig. 5). For example, the reduction in aerosol burden is
much larger in central-eastern China during the SD El Niño events than during the LD
El Niño events, with maximum negative anomalies, respectively, reaching -1.6 and
-0.6 $mg\ m^{-2}$. Overall, SD El Niño events yield stronger impacts on aerosol pollution in
China than LD El Niño events, especially in central-eastern China with negative
pollution anomalies.
**3.2 Mechanisms of SD and LD El Niño impacts on aerosols**
Since aerosols and their precursor gas emissions are prescribed at the same rates
in the control (CLIM) and SD/LD sensitivity simulations, changes in meteorological
factors such as circulation and precipitation play dominant roles in altering aerosol
concentrations by influencing the regional transport and wet removal of aerosols
(Yang et al., 2016a). Previous studies also suggested that aerosol variations during
ENSO events were controlled by ENSO-related circulation and precipitation changes



(Yu et al., 2019; Zhao et al., 2018). Here, we examine the atmospheric circulation and
precipitation anomalies and the associated aerosol processes during the SD and LD El
Niño to explore the mechanisms of the two types of El Niño effects on aerosols in
China.
Both the SD and LD El Niño events trigger negative anomalies in sea level
pressure (SLP) over the eastern China and East China Sea and positive anomalies
over the Philippine Sea and Sea of Okhotsk (not shown), leading to anomalous
cyclonic and anticyclonic circulations over these regions, respectively (Figs. 7a and
7b). At 850 hPa, the anomalous cyclonic circulation over the East China Sea causes
anomalous northerly winds over central-eastern China, enhancing the prevailing
northwesterly winds in winter. The enhanced winds favor the aerosol dispersion,
which explains the decrease in PM$_{2.5}$ concentrations over central-eastern China during
El Niño events relative to the climatological mean. In addition, at 500 hPa, most areas
over China have an anomalous low pressure (Figs. 7d and 7e), which increases the
atmospheric instability and strengthens the aerosol vertical mixing and dispersion
over central-eastern China. Over southern China, the aerosol variations are
significantly affected by the regional transport of particles from South and Southeast
Asia. During El Niño events, anomalous southwesterly winds at the northwest edge of
the anomalous anticyclone over the Philippine Sea bring aerosols from South and
Southeast Asia to southern China, contributing to the aerosol increases in southern
China relative to the climatological mean (Figs. 7a and 7b). In the northeastern China,
anomalous southeasterly winds associated with the anomalous anticyclonic circulation
over Sea of Okhotsk weaken the wintertime prevailing northwesterly winds, giving
rise to the aerosol increases in the northeastern China during El Niño events. In
addition, the anomalous anticyclone brings aerosols from South and Southeast Asia to
northeastern China that will be discussed next, contributing to the aerosol pollution in
northeastern China during El Niño events.
Compared to LD El Niño events, the negative anomaly of SLP over the East
China Sea during the SD El Niño events is stronger and extends deeply into the
central-eastern China, resulting in anomalous northerly winds over central-eastern
China and southeasterly winds over northeastern China in the lower atmosphere (Fig.
7c). The wind anomalies intensify the aerosol dispersion in central-eastern and
accumulation in northeastern China, leading to a stronger effect of El Niño with short
duration on the aerosol variation in China. Furthermore, the anomalous northerly
winds in both lower atmosphere and 500 hPa over southern China (Figs. 7c and 7f)
are unconducive to the regional transport of aerosols from South and Southeast Asia
to central-eastern China.
In addition to the regional transport prompted by anomalous atmospheric
circulations, El Niño can influence aerosol wet removal through perturbing
precipitation. As described in Figs. 8a and 8b, the spatial patterns of winter
precipitation anomalies in China during SD and LD El Niño events are similar, with
positive anomalies located along the southeastern coastal areas due to the additional
moisture transport by anomalous southwesterly winds over the South China Sea.
However, the two types of El Niño events differ in the magnitude of precipitation
anomalies. In central-eastern China, precipitation decreases during SD El Niño events,
compared to LD El Niño events, whereas precipitation increases over eastern coastal
areas and northeastern China (Fig. 8c). This is linked to the anomalous cyclonic
circulation over central-eastern China (Fig. 7c), which hinders moisture from South
China Sea to central-eastern China but brings in moisture from Sea of Japan to
northeastern China. Over Pearl River Delta, precipitation decreases during SD El
Niño events but increases during LD El Niño events, which is also associated with the
anomalous northerly winds and corresponding impact on moisture transport over this
region. In general, aerosol wet deposition decreases in central-eastern China and
increases over southern and northeastern China during El Niño events (Figs. 8d and
8e). With short duration but strong intensity, El Niño events have larger impacts on
aerosol wet removal than those with long duration (Fig. 8f). However, the wet
removal shows a positive relationship with the aerosol concentration, which should be
a negative relationship in theory if other conditions remain unchanged. Therefore, the
differences in aerosols triggered by El Niño events with different durations are
primarily due to the impact of changes in atmospheric circulation on the accumulation
and transport of aerosols rather than the impact of precipitation on aerosol removal.
Both the accumulation/dispersion of local aerosols within China and regional
transport of aerosols from South and Southeast Asia can contribute to the aerosol
changes in China during El Niño events. With emissions of aerosols and precursor
gases in South and Southeast Asia turned off, the decrease pattern of $PM_{2.5}$ over
central-eastern China does not change (Fig. 9), suggesting that
accumulation/dispersion of local aerosols dominates the aerosol change over this
region during El Niño events. Over southern China, the increase of $PM_{2.5}$ burden is



weakened when the South and Southeast Asian emissions are turned off, indicating
that regional transport of aerosols from South and Southeast Asia have a large
contribution to the aerosol variation over this region. It is interesting that, without
emissions from South and Southeast Asia, both near-surface concentration and
column burden of $PM_{2.5}$ in northeastern China decrease during El Niño events relative
to the climatological mean, but the change reverses to increase when the South and
Southeast Asian emissions are considered. It indicates that the aerosol enhancements
in northeastern China during the El Niño events are most likely influenced by aerosol
transport from South and Southeast Asia due to anomalous southeasterly winds at the
eastern edge of the anomalous cyclonic circulation in eastern China (Fig.7c), which
warrants further analysis using a source-receptor model such as CAM5-EAST (Ren et
al., 2020; Yang et al., 2020).

## 3.3 Quantitative impacts on regional $PM_{2.5}$ concentrations and haze days

Table 1 summarizes the simulated regional mean $PM_{2.5}$ concentrations and
number of haze days in DJF over the sub-regions in China (Fig. 10), including the
North China Plain (NCP, 35−41°N, 114−120°E), Sichuan Basin (SCB, 28−33°N,
103−108°E), Yangtze River Delta (YRD, 29−34°N, 118−121.5°E), Pearl River Delta
(PRD, 21.5−25°N, 111−116°E), Northeast Plain (NEP, 41−48°N, 120−130°E), the
Yunnan-Guizhou Plateau (YGP, 23−27°N, 100−110°E), and the Fenwei Plain (FWP,
33−35°N, 106−112°E and 35−38°N, 110−114°E) from CLIM, SD and LD simulations.
Haze days are defined as days with daily near-surface $PM_{2.5}$ concentrations above the
90[th] percentile of the CLIM $PM_{2.5}$ concentrations in each sub-region of China.
During El Niño events, DJF mean near-surface $PM_{2.5}$ concentrations decrease
over NCP, SCB, YGP, and FWP regions and increase over PRD and NEP in both SD
and LD, compared to CLIM. Although the $PM_{2.5}$ concentrations show an increase in
SD and a decrease in LD over YRD region, the changes are statistically insignificant
in this region (Fig. 5). SD El Niño events have a stronger modulation on aerosols in
China than LD El Niño events. Over the regions with concentration decreases (NCP,
SCB, YGP, and FWP), regional mean near-surface $PM_{2.5}$ concentration in LD is lower
than CLIM by 0.24 μg m$^{-3}$, while the reduction reaches 1.22 μg m$^{-3}$ in SD, about 5
times as that of LD. Over the regions with concentration increases (PRD and NEP),
the $PM_{2.5}$ increase in SD relative to CLIM is 0.74 μg m$^{-3}$, which is also higher than
the 0.56 μg m$^{-3}$ in LD.





Similar to the PM$_{2.5}$ concentration, the modulation of SD El Niño events on haze

days are 2-3 times as high as that of LD El Niño events. During LD El Niño events,
the number of haze days in DJF at NCP, SCB and FWP is reduced by 1.14, 0.73 and
1.53 days, respectively, compared to the climatological mean, while the decrease in
haze days during SD El Niño events is more substantial (1.87, 2.13 and 2.87 days).
The probability density distributions of PM$_{2.5}$ concentrations over NCP, SCB and
FWP in SD and LD also shift to the left, relative to CLIM (Fig. 11). Consistent with
the stronger modulation of SD El Niño events discussed above, the shift in SD is more
than that in the LD simulation. In addition, YRD, PRD and NEP regions all have
increases in haze days in DJF during SD and LD, relative to CLIM. Similarly, during
SD and LD El Niño events, the probability density distributions of high values of
PM$_{2.5}$ concentrations over YRD, PRD and NEP slightly shift to the right relative to
CLIM. The number of haze days in DJF over YGP decreases during SD El Niño
events by 1.4 days, but there is a slight increase of 0.4 days during LD El Niño events,
likely due to the opposite aerosol changes in the eastern and western parts of YGP
region (Fig. 5). There are more (fewer) haze days in both SD and LD than in CLIM
over YRD, PRD and NEP (NCP, SCB and FWP), which is inconsistent with the
simulated greater (less) precipitation over these regions caused by El Niño events. It
further indicates that anomalies in precipitation are not the most dominant factor
modulating winter haze days in China during El Niño events, but rather the
anomalous aerosol accumulation/dispersion and transport due to anomalous
atmospheric circulation.

## 441   3.4 Historical increase in SD El Niño events

Many studies have suggested an increase in the variability of El Niño events

under greenhouse warming (Cai et al., 2018; Grothe et al., 2020). However, few
studies have shown the historical changes in El Niño with different durations, which
would further impact aerosol concentrations and haze days in China.

Here we show the occurrence of SD and LD El Niño events since the

preindustrial era in Fig. 12. The number of SD El Niño events fluctuated but has
increased significantly during the past few decades, especially after 1940s. The
occurrence of SD El Niño increased from one event per fifteen years during
1941–1955 to four events per fifteen years during 2001–2015, with the increase at
confidence level of 92%, while LD El Niño events did not present a significant trend
in the historical period. Wu et al. (2019) found that the duration of El Niño is mainly





influenced by the timing of onset, associated with the early onset of delayed negative
oceanic feedback as well as the fast adjustments of the tropical Indian and Atlantic
Oceans to the tropical Pacific Ocean warming. It is conjectured that the onset timing
of El Niño events gets earlier under greenhouse forcing, but the detailed analysis is
out of the scope of this study. Nevertheless, because the frequency of the El Niño
events with short duration increased significantly, the modulation by El Niño events
on wintertime aerosols in China has intensified in the past few decades.

## 4. Conclusion and discussions
As a prominent climate phenomenon, El Niño triggers atmospheric circulation
and precipitation anomalies on a global scale, thus having important effects on haze
days and aerosol pollution in China. In this study, the impacts of different temporal
types of El Niño events with short and long duration on aerosols in China are
simulated using the state-of-the-science E3SM model.
For both SD and LD El Niño events, their changes to the DJF mean $PM_{2.5}$
concentrations have similar spatial distributions over China, relative to the
climatological mean. The anomalous anticyclonic circulation over the Sea of Okhotsk
weakens the prevailing northwesterly winds in DJF in northeastern China and
enhances the accumulation of locally emitted aerosols, along with the anomalous
southeasterly winds at the eastern edge of the anomalous cyclonic circulation in
eastern China that intensifies the aerosol transport from South and Southeast Asia to
northeastern China. The near-surface $PM_{2.5}$ concentration in northeastern China
increases by 1-2 $\mu$g m$^{-3}$ during El Niño events relative to the climatological conditions.
In southern China, the anomalous anticyclonic circulation over the Philippine Sea
facilitates the transport of aerosols from South and Southeast Asia to southern China
and thus the near-surface $PM_{2.5}$ concentrations in southern China increase by 1-2 $\mu$g
m$^{-3}$. The decrease in near-surface $PM_{2.5}$ concentrations in central-eastern China is
mainly controlled by the enhanced northerly winds from the anomalous cyclonic
circulation over eastern China and the East China Sea, leading to the dispersion of
local aerosols, while precipitation change has little effect on aerosols here. Compared
to LD El Niño events, due to the anomalous cyclonic circulation over eastern China,
SD El Niño events exhibit a stronger reduction (1-2 $\mu$g m$^{-3}$) in near-surface $PM_{2.5}$
concentrations over central-eastern China and a larger increase (0.6 $\mu$g m$^{-3}$) in



northeastern China. Overall, El Niño with short duration has a stronger modulation on
wintertime aerosols in China than El Niño with long duration.
Compared with CLIM, mean near-surface $PM_{2.5}$ concentrations in DJF decrease
over NCP, SCB, YKP and FWP regions and increase over PRD and NEP in both SD
and LD, but the decrease over these regions in SD El Niño events reaches 1.22 μg m$^{-3}$,
about 5 times as large as that of LD. Similarly, both SD and LD El Niño events induce
less (more) haze days in DJF than CLIM over NCP, SCB and FWP (YRD, PRD and
NEP). However, the decreases in haze days in DJF at NCP, SCB and FWP during SD
El Niño events are 2-3 times more than that during LD El Niño events.
We also found that the occurrence frequency of SD El Niño events increased
from one event per fifteen years during 1941–1955 to four events per fifteen years
during 2001–2015, whereas LD El Niño events did not exhibit a significant trend in
the historical period. In particular, seven SD El Niño events have occurred since the
1990s, but no LD El Niño event occurred. Compared to LD El Niño events, SD El
Niño events have a greater impact on wintertime aerosols over China. Therefore, the
impact of El Niño events on wintertime aerosols in China has intensified in the past
few decades due to their short durations.
Our results of the important effect of SD El Niño events and its recent
intensification are of great significance for the understanding of El Niño on China's
haze pollution, alleviating air pollution, and coping with climate change. The
simulated spatial patterns of aerosol changes during El Niño events resemble those in
previous studies (Feng et al., 2016; Yu et al., 2019; Zhao et al., 2018). However, there
are still some inadequacies remaining to be improved. Natural aerosols including dust
and sea salt were not considered in this study. The EAMv1 model largely
underestimated $PM_{2.5}$ concentration in China related to the lack of nitrate and
ammonium aerosols and other model biases. We also found that, during El Niño
events, more aerosols from South and Southeast Asia can be transported to
northeastern China, leading to an increase in aerosol concentrations over there. Thus,
more in-depth analysis is needed in future studies. In addition, during the cooling
phase of ENSO, La Niña events may also have various durations and can have
different impacts on air pollutions in China, which merits further investigation.






*Data availability*
The E3SMv1 model is available at https://e3sm.org/ (last access: 1 February 2021).
Our results can be made available upon request. Hourly observations of $PM_{2.5}$
concentrations over China can be derived from the China National Environmental
Monitoring Centre (http://www.cnemc.cn, last access: 1 February 2021)

*Author contributions*
YY designed the research. LZ performed the model simulations and analyzed the data.
All the authors discussed the results and wrote the paper.

*Competing interests*
The authors declare that they have no conflict of interest.

*Acknowledgments*
This study was supported by the National Natural Science Foundation of China (grant
41975159) and the National Key Research and Development Program of China (grant
2020YFA0607803 and 2019YFA0606800). HW acknowledges the support by the U.S.
Department of Energy (DOE), Office of Science, Office of Biological and
Environmental Research (BER), as part of the Earth and Environmental System
Modeling program. The Pacific Northwest National Laboratory (PNNL) is operated
for DOE by the Battelle Memorial Institute under contract DE-AC05-76RLO1830.






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



**Table 1.** The seasonal mean aerosol concentrations (unit: μg m$^{-3}$) and number of haze days (unit:
day) in December-January-February (DJF) over various regions of China, including NCP, SCB,
YRD, PRD, NEP, YGP, and FWP from CLIM, SD and LD experiments. Haze days are defined as
days with daily average near-surface PM$_{2.5}$ concentrations above the 90$^{th}$ percentile in each region.
The values in brackets represent the anomalies in SD and LD relative to CLIM.

|  |  | NCP | SCB | YRD | PRD | NEP | YGP | FWP |
|---|---|---|---|---|---|---|---|---|
| **Mean Conc.** | CLIM | 24.87 | 32.33 | 27.98 | 17.26 | 9.42 | 20.19 | 25.11 |
|  | SD | 23.73 (-1.14) | 31.16 (-1.17) | 28.21 (+0.23) | 18.20 (+0.94) | 9.95 (+0.53) | 19.55 (-0.64) | 23.17 (-1.94) |
|  | LD | 24.76 (-0.11) | 32.09 (-0.24) | 27.84 (-0.14) | 18.09 (+0.83) | 9.71 (+0.29) | 20.07 (-0.12) | 24.64 (-0.47) |
| **Haze Days** | CLIM | 9.00 | 9.00 | 9.00 | 9.00 | 9.00 | 9.00 | 9.00 |
|  | SD | 7.20 (-1.87) | 6.87 (-2.13) | 9.87 (+0.87) | 10.27 (+1.27) | 10.87 (+1.80) | 7.60 (-1.40) | 6.13 (-2.87) |
|  | LD | 7.93 (-1.14) | 8.27 (-0.73) | 9.53 (+0.53) | 11.07 (+2.07) | 10.00 (+0.93) | 9.40 (+0.40) | 7.47 (-1.53) |






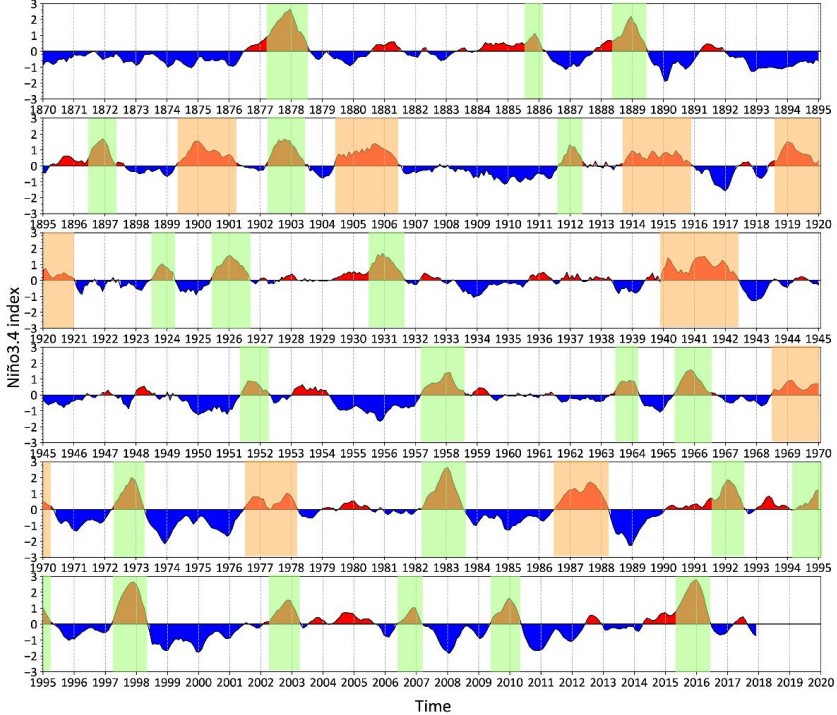


**Figure 1.** Time series of the Niño3.4 index (°C) based on the merged Hadley-NOAA/OI SST
dataset for 1870-2017.The time series were detrended and smoothed with a 3-month running
average filter. Highlighted slots illustrate the SD (green) and LD (orange) El Niño events.





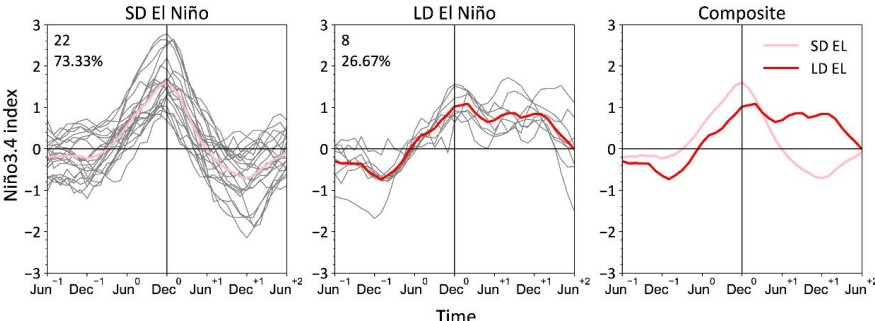



**Figure 2.** Time series of the Niño3.4 index (℃) overlaid from Jun[-1] to Jun[+2] for (left) SD and
(middle) LD El Niño events during 1870-2017. The individual and composite events are shown by
thin gray and bold red curves, respectively. The total number and percentage of events are shown
at the upper left corner of each panel. A comparison of the composite time series of Niño3.4 index
for SD and LD events is shown in the right panel.

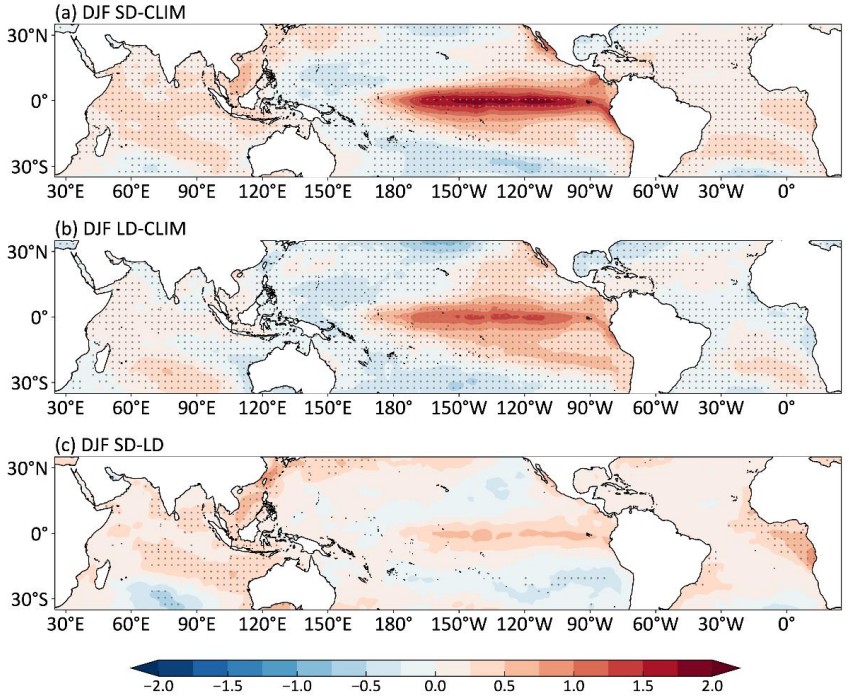

**Figure 3.** Composite differences in DJF mean SST (°C) between SD (a) / LD (b) El Niño events and climatological mean over 1870-2017 and between SD and LD (c) El Niño events. Differences that are statistically significant at 95% from a two-tailed T-test are stippled.


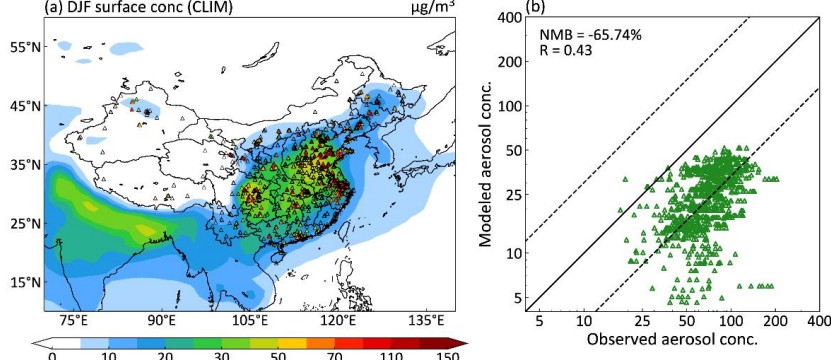



**Figure 4.** Spatial distributions (a) and scatter plots (b) of observed and simulated DJF mean
near-surface PM$_{2.5}$ concentrations (µg m$^{-3}$) from the CLIM experiment. Solid line represents 1:1
ratio and dashed lines mark 1:3 and 3:1 ratios. The observed concentrations are derived from the
CNEMC in December 2014-February 2015. The normalized mean deviation (NMB) and the
correlation coefficient (R) between observations and simulation are shown in the upper left corner
of the right panel. NMB = 100% × $\sum (M_i - O_i) / \sum O_i$ , where $M_i$ and $O_i$ are the simulated and
observed values at the site i, respectively.




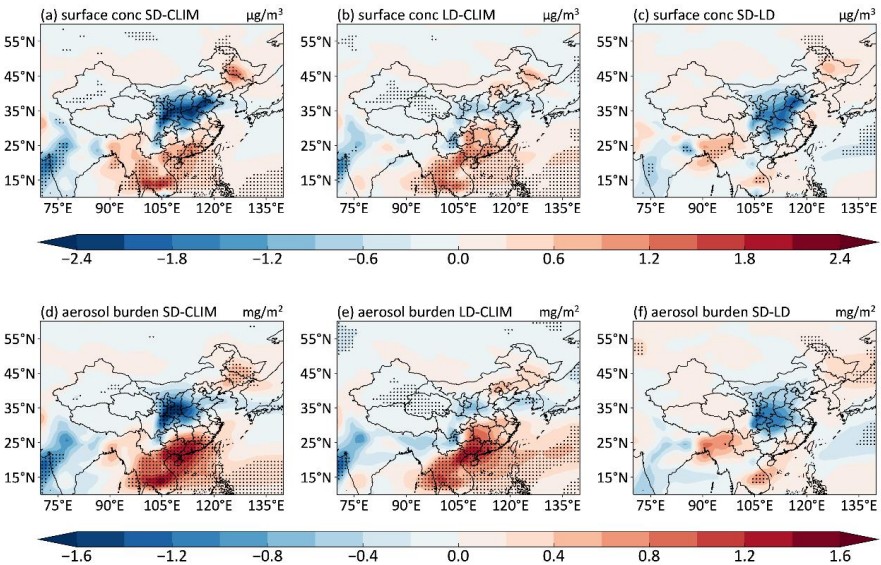

**Figure 5.** Composite differences in DJF mean near-surface PM$_{2.5}$ concentrations (µg m$^{-3}$) and aerosol column burdens (mg m$^{-2}$) between SD and CLIM (a, d), LD and CLIM (b, e), and SD and LD (c, f). The stippled areas indicate statistical significance with 90% confidence from a two-tailed T-test.

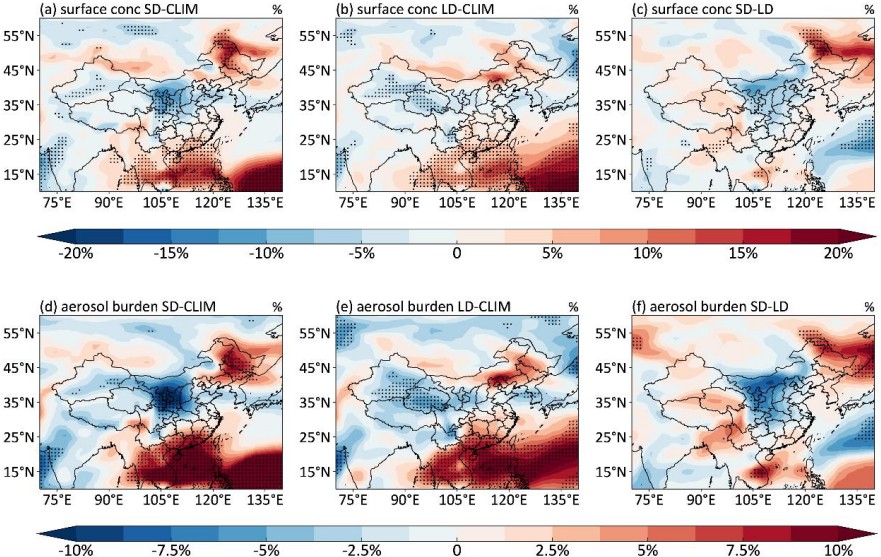

**Figure 6.** Composite differences (%) in DJF mean near-surface $PM_{2.5}$ concentrations and aerosol column burdens between SD and CLIM (a, d), LD and CLIM (b, e), and SD and LD (c, f), relative to CLIM. The stippled areas indicate statistical significance with 90% confidence from a two-tailed T-test.




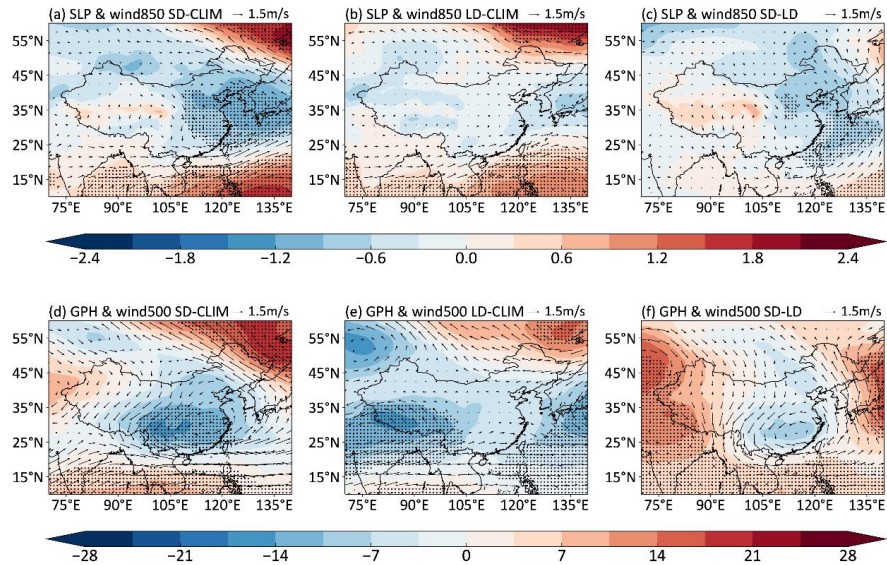



**Figure 7.** Composite differences in DJF mean sea level pressure (SLP, shaded; units: hPa) and
wind at 850 hPa (WIND850, vector; units: m s[-1]) (top panels) and geopotential height at 500 hPa
(GPH500, shaded; units: m) and wind at 500 hPa (WIND500, vector; units: m s[-1]) (bottom panels)
between SD and CLIM (a, d), LD and CLIM (b, e), and SD and LD (c, f). The stippled areas
indicate statistical significance with 90% confidence from a two-tailed T-test.




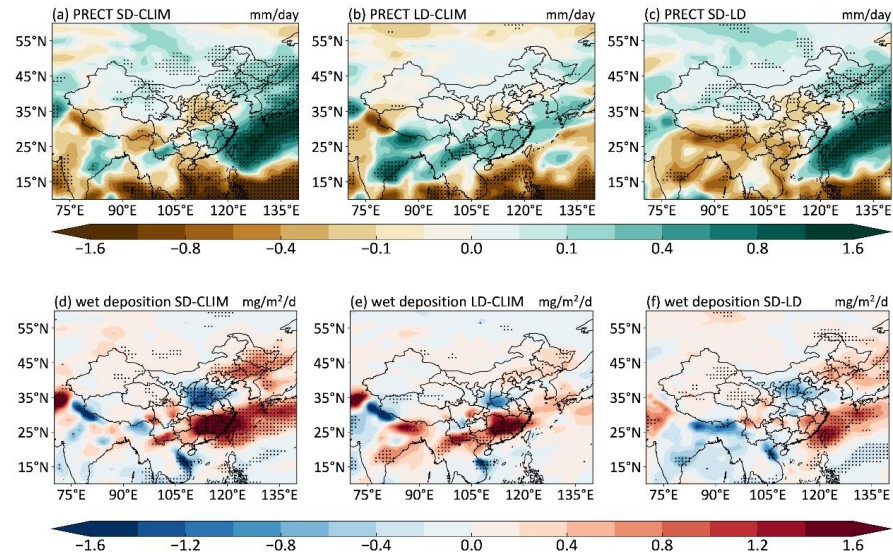



**Figure 8.** Composite differences in DJF mean precipitation rate (top panels; units: mm day$^{-1}$) and
wet deposition of PM$_{2.5}$ (bottom panels; units: mg m$^{-2}$ d$^{-1}$) between SD and CLIM (a, d), LD and
CLIM (b, e), and SD and LD (c, f). The stippled areas indicate statistical significance with 90%
confidence from a two-tailed T-test.



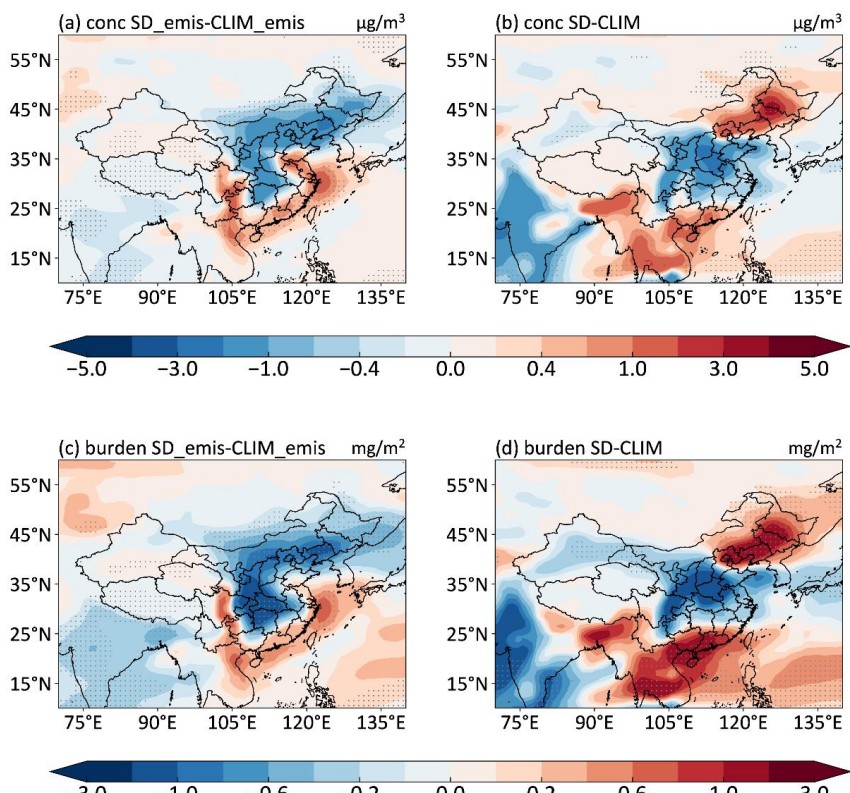


**Figure 9.** Composite differences in DJF mean near-surface PM$_{2.5}$ concentration (μg m$^{-3}$) and
aerosol column burden (mg m$^{-2}$) between SD_emis and CLIM_emis (a, c) SD and CLIM (b, d).
The stippled areas indicate statistical significance with 90% confidence from a two-tailed T-test.






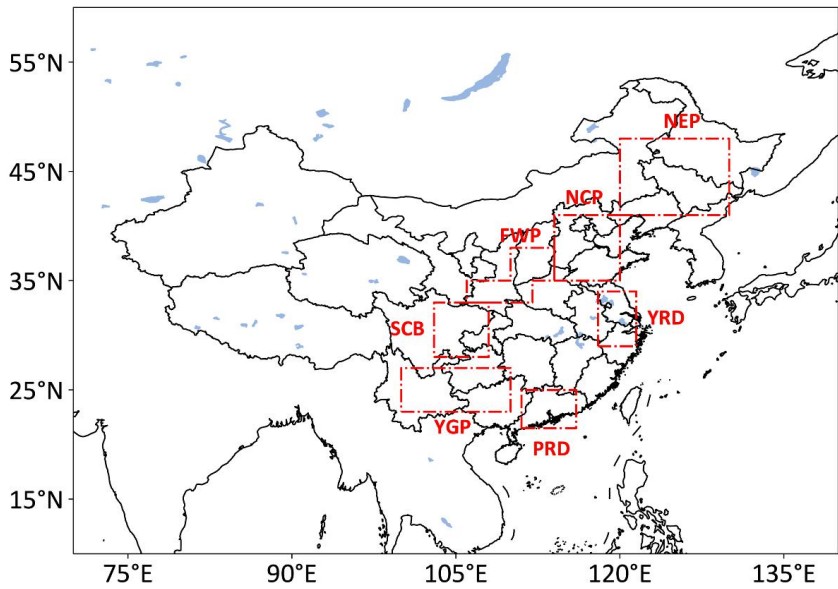


**Figure 10.** Subregions of China defined in this study, including the North China Plain (NCP,
35−41°N, 114−120°E), the Sichuan Basin (SCB, 28−33°N, 103−108°E), the Yangtze River Delta
(YRD, 29−34°N, 118−121.5°E), the Pearl River Delta (PRD, 21.5−25°N, 111−116°E), the
Northeast Plain (NEP, 41−48°N, 120−130°E), the Yunnan-Guizhou Plateau (YGP, 23−27°N,
100−110°E), and the Fenwei Plain (FWP, 33−35°N, 106−112°E and 35−38°N, 110−114°E).






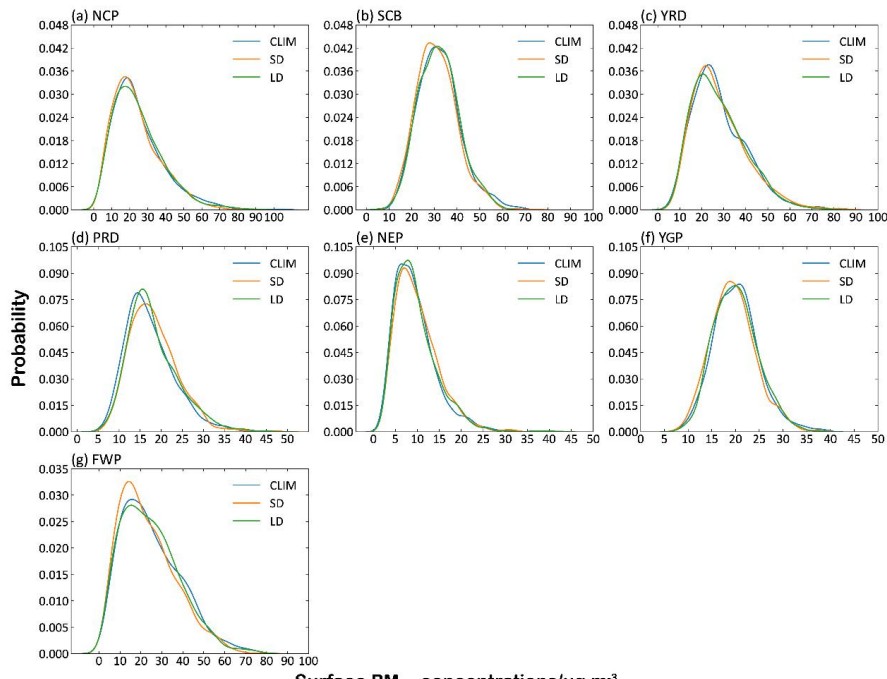



**Figure 11.** Probability density distributions of daily near-surface PM$_{2.5}$ concentrations (µg m$^{-3}$) in DJF over various subregions of China.







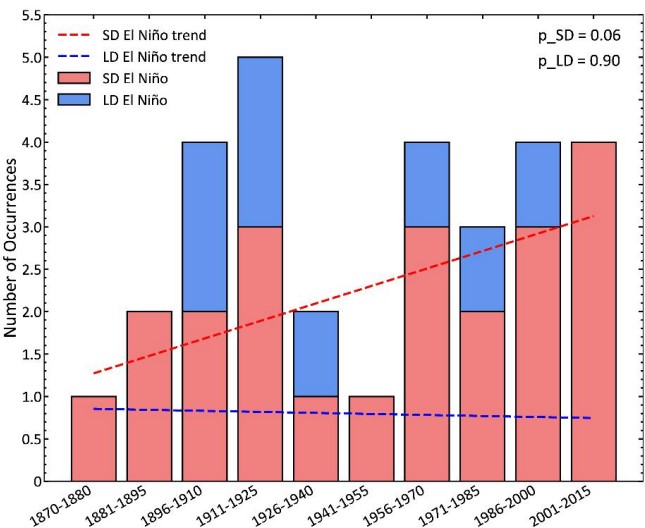

**Figure 12.** Stacked histograms of the number of SD and LD El Niño events per 15 years (except 1870-1880 for 10 years) during 1870-2015.The red and blue dashed lines indicate linear trends in the number of SD and LD El Niño events, respectively. Their p-values are shown in the upper right corner of the figure, which indicate the increasing trend of SD at a two-tailed T-test confidence level of 94% for 1870-2015 (87% for 1880-2015 and 92% for 1940-2015) statistical significance.