# Peer review of "Intensified modulation of aerosol pollution in China by El"

_Atmospheric Chemistry and Physics, 2021_

## Author Comment (AC1)

**Manuscript # acp-2021-166**

**Responses to Referee #1**

*This study investigated the different impacts of short duration (SD) and long duration (LD) El Niño on the aerosol pollution in China. The authors found that SD El Niño exerts more significant influences on the $PM_{2.5}$ concentrations in winter, which increase in northeastern and southern China and decrease in central-eastern China. The anomalous atmospheric circulation induced by SD El Niño is the dominant reason for $PM_{2.5}$ concentration changes. These findings are interesting and useful to air quality prediction and improvement. The logic of this article is good. However, there are some major concerns about the physical mechanism that I doubt. The comments below that I expect are helpful for improving the manuscript.*

We thank the reviewer for all the insightful comments. Below, please see our point-by-point response (in blue) to the specific comments and suggestions and the changes that have been made to the manuscript, in an effort to take into account all the comments raised here.

*Major comments:*

*1. The anomalous atmospheric circulation induced by El Niño plays a vital role in changing the $PM_{2.5}$ concentration distribution. The authors should give the observed atmospheric circulation conditions induced by SD and LD El Niño using the reanalysis data. Then, the authors can compare the observed atmospheric circulation with the simulated circulation, and can further evaluate whether the model simulates a reasonable atmospheric circulation. This check is very important, because atmospheric circulation induced by El Niño determines the conclusion of this study.*

Thanks for the suggestion. We have added Fig. 7 in order to compare the observed with the simulated atmospheric circulation anomalies during SD and LD El Niño relative to the climatology. "To further verify the model simulations in capturing atmospheric circulation anomalies during SD and LD El Niño events, the wind fields are compared with those from ERA5 reanalysis data. The anomalous atmospheric circulation patterns in the latest SD El Niño event (2015/2016) and LD El Niño event (1986/1987) relative to the climatological mean (1950-2017) from the ERA5 are shown in Fig. 7. Overall, the SD and LD El Niño-induced anomalous atmospheric circulations over China simulated in E3SM are in consistent with the reanalysis data. Both of them show the anomalous northerly winds over central-eastern China at 850 hPa during SD El Niño compare to LD El Niño. In

addition, obvious anomalous cyclone at 500 hPa over most of China can be seen in both E3SM and ERA5." We have added these descriptions in the revised manuscript.

[Figure]

**Figure 6.** Composite differences in DJF mean sea level pressure (SLP, shaded; units: hPa) and wind at 850 hPa (WIND850, vector; units: m s-1) (top panels) and geopotential height at 500 hPa (GPH500, shaded; units: m) and wind at 500 hPa (WIND500, vector; units: m s-1) (bottom panels) between SD and CLIM (a, d), LD and CLIM (b, e), and SD and LD (c, f). The stippled areas indicate statistical significance with 90% confidence from a two-tailed T-test.

[Figure]

Figure 7. Composite differences in DJF mean winds at 850 hPa (m s⁻¹) (top panels) and 500 hPa (m

s$^{-1}$) (bottom panels) between 2015/2016 SD El Niño and climatological mean (1950-2017) (a, d), 1986/1987 LD El Niño and climatological mean (b, e), and 2015/2016 SD El Niño and 1986/1987 LD El Niño (c, f) from the EAR5 reanalysis data. The data were detrended over 1950-2017.

2. It is known that the anticyclone over the western North Pacific is the key atmospheric circulation system that El Niño exerts its impact on East Asia. Certainly, the southeasterly winds on the western side of this anticyclone lead to increase of the PM$_{2.5}$ concentration. However, the authors reported that the decrease of the PM$_{2.5}$ concentration in central-eastern China is attributed to the anomalous northerly winds of the cyclone over the East China Sea (Figure 7a). This anomalous cyclone is rarely reported. I doubt whether this anomalous cyclone indeed exists in observation, or it only appears in simulation? So, I suggest the authors should check the observational circulation condition using the reanalysis data to verify the simulated result.

Thanks for your insightful suggestion. As we replied above, we compared the atmospheric circulation anomalies produced by E3SM with ERA5 reanalysis data. Both of them showed anomalous northerly winds of an anomalous cyclone over East China Sea during SD El Niño events relative to the climatology. Chen et al. (2018) also found that an anomalous cyclone appeared over the western North Pacific in late 1986 El Niño. So, the anomalous cyclone does exist in observation. However, we note that atmospheric circulation in the real world is influenced not only by El Niño but also by many other climate phenomenon, such as Arctic Oscillation and Pacific Decadal Oscillation, while the E3SM simulation in this study focuses on the pure effects of El Niño. Since that El Niño is a climate phenomenon in the equatorial Pacific Ocean and has less impact over high latitudes, the circulation anomalies produced by E3SM differ with observations in high latitudes, although it is not the focus area of this study.

Minor comments:
3. Please pay attention to the singular and plural in English grammar, for example:
Line 45, "have" should be changed into "has"
Revised.

Line 77, "have" should be changed into "has"
Revised.

Line 170, "is" should be changed into "are"
Revised.

*Line 188, "are" should be changed into "is"*

Revised.

*4. Line 172-176, please rewrite this sentence*

We have revised the sentence as follows: "Niño 3.4 index is detrended SST anomaly over the Niño 3.4 region (170°W-120°W, 5°S-5°N). El Niño event is firstly identified when a 3-month running mean Niño 3.4 index is greater than 0.75°C in any month from Oct[0] to Feb[1] of its developing phase. If the Niño 3.4 index is higher than 0.5°C in any month from Oct[1] to Feb[2] of its decaying phase, the El Niño event is an LD El Niño event; otherwise, it is an SD El Niño event (Wu et al., 2019)."

*5. Line 191, plus "anomaly" after "SST"*

Revised.

Reference:

Chen, M., and Li, T.: Why 1986 El Niño and 2005 La Niña evolved different from a typical El Niño and La Niña, Clim. Dyn., 51, 4309–4327, https://doi.org/10.1007/s00382-017-3852-1, 2018.

---

## Author Comment (AC2)

**Manuscript # acp-2021-166**

**Responses to Referee #2**

*As mentioned in the review of this article, many scholars have conducted the relevant studies on the impact of El Niño events on winter aerosol pollution over China. However, there are some uncertainties in those studies and thus it is necessary to continue to carry out In-depth research to reduce the uncertainty. The ingenuity of this research is that the classification method is distinguished according to the life cycle (i.e. SD and LD) method instead of the traditional El Niño such as intensity, CP or EP of the El Niño. So this study is very meaningful, but it can still be further improved in the following aspects.*

We thank the reviewer for all the insightful comments. Below, please see our point-by-point response (in blue) to the specific comments and suggestions and the changes that have been made to the manuscript, in an effort to take into account all the comments raised here.

*1. The author can try to discuss the relationship and difference between the SD and LD classification methods and the intensity classification in previous studies or the CP and EP classification.*

The intensity classification focuses on the strength of an El Niño event. The East Pacific (EP) and Central Pacific (CP) types of El Niño are classified on the basis of the spatial location of sea surface temperature (SST) anomaly. The SD and LD classification is based on the duration of an El Niño event. These are briefly described in Section 1.

*2. Although the authors focused on the analysis of events in different seasons throughout the year when distinguishing between the SD and LD types of El Niño, when studying the impact of different types of El Niño on aerosol pollution, DJF is selected as the researched season, so I recommend the title of this article should be revised to **Intensified modulation of winter aerosol pollution in China by El Niño with short duration**.*

Thanks for the suggestion. We have revised the title of our manuscript to *Intensified modulation of winter aerosol pollution in China by El Niño with short duration.*

*3. Previous study has shown (Sun et al., 2018) that GCM models have certain limitations to capture climate anomalies generated by El Niño, so there are usually some inconsistencies between the*

*simulation results and the observation results. It is recommended that the author face up to these problems and discuss the uncertainty of the study.*

*Reference:*

*Sun, J., Li, H., Zhang, W., Li, T., Zhao, W., Zuo, Z., Guo, S., Wu, D., and Fan, S.: Modulation of the ENSO on Winter Aerosol Pollution in the Eastern Region of China, J. Geophys. Res. Atmos.,123, 11,952-11, https://doi.org/10.1029/2018jd028534, 2018.*

Thanks for your suggestion. We have added Fig. 7 in order to compare the observed with the simulated atmospheric circulation anomalies during SD and LD El Niño relative to the climatology. "To further verify the model simulations in capturing atmospheric circulation anomalies during SD and LD El Niño events, the wind fields are compared with those from ERA5 reanalysis data. The anomalous atmospheric circulation patterns in the latest SD El Niño event (2015/2016) and LD El Niño event (1986/1987) relative to the climatological mean (1950-2017) from the ERA5 are shown in Fig. 7. Overall, the SD and LD El Niño-induced anomalous atmospheric circulations over China simulated in E3SM are in consistent with the reanalysis data. Both of them show the anomalous northerly winds over central-eastern China at 850 hPa during SD El Niño compare to LD El Niño. In addition, obvious anomalous cyclone at 500 hPa over most of China can be seen in both E3SM and ERA5." We have added these descriptions in the revised manuscript.

[Figure]

**Figure 6.** Composite differences in DJF mean sea level pressure (SLP, shaded; units: hPa) and wind at 850 hPa (WIND850, vector; units: m s-1) (top panels) and geopotential height at 500 hPa

(GPH500, shaded; units: m) and wind at 500 hPa (WIND500, vector; units: m s-1) (bottom panels) between SD and CLIM (a, d), LD and CLIM (b, e), and SD and LD (c, f). The stippled areas indicate statistical significance with 90% confidence from a two-tailed T-test.

[Figure]

Figure 7. Composite differences in DJF mean winds at 850 hPa (m s$^{-1}$) (top panels) and 500 hPa (m s$^{-1}$) (bottom panels) between 2015/2016 SD El Niño and climatological mean (1950-2017) (a, d), 1986/1987 LD El Niño and climatological mean (b, e), and 2015/2016 SD El Niño and 1986/1987 LD El Niño (c, f) from the EAR5 reanalysis data. The data were detrended over 1950-2017.

---

## Author Comment (AC3)

**Manuscript # acp-2021-166**

**Responses to Referee #3**

The El Niño (ENSO) is an irregular periodic variation in sea surface temperature in the tropical Pacific Ocean, affecting the Walker Circulation and displacing the convective area. It can lead to significant anomalies in atmospheric general circulations and weather conditions, which may further impact on haze pollutions in China. In this article, authors used the state-of-the-science Energy Exascale Earth System Model version 1 (E3SMv1) to estimate the impacts of El Niño with short and long durations on PM2.5 during winter in China. They further investigated underlying mechanisms of PM2.5 variations controlled by ENSO-related circulation and precipitation changes. The results are helpful to improve the understanding of modulation of aerosol pollution in China by El Niño over the study region. I think the manuscript can be accepted after the following concerns are addressed.

We thank the reviewer for all the insightful comments. Below, please see our point-by-point response (in blue) to the specific comments and suggestions and the changes that have been made to the manuscript, in an effort to take into account all the comments raised here.

**General Comments:**

1. The Introduction should be more concise. The authors should work further to reduce the main text to exclude any unnecessary contents like "clean air actions", and explain how spatiotemporal variations of aerosol in China controlled by ENSO-related changes in not only in wind speed and precipitation, but Walker Circulation and convections as well in more detail.

Thanks for the suggestion. We have pruned the introduction as follows: "In order to alleviate air pollution, a comprehensive and better scientific understanding of all factors that can affect aerosol concentrations and haze pollution in China is required." We agree with the reviewer that spatiotemporal variations of aerosols in China are also controlled by atmospheric circulation, which we have presented in the introduction. However, the ENSO-related changes in Walker circulation and the associated convections are more related to the aerosols over the tropics. Because we focused on aerosols in China, we did not discuss these factors in the manuscript.

2. The authors intended to show the difference of El Niño with short and long durations in section 2.2, where the background climates are quite different. Thus more discussion on their difference in climatological means (e.g. general circulation, temperature, precipitation, humidity and wind) are recommend.

In section 2.2, we only showed the definition of SD and LD El Niño. The differences in climatology have been discussed in the results in details when analyzing their impact of aerosol distributions in China.

**Minor suggestions:**

1. Line 92, "Based on haze day counting mainly using atmospheric visibility, many studies found" may be replaced with "Many studies counted haze days based on atmospheric visibility and found". Revised.

2. Line 96, "several studies ... from satellite retrievals". Insert some references there.
We have inserted the following references after the sentence: "several studies ... from satellite retrievals (Jeoung et al., 2014; Sun et al., 2018)".

**3. Line 141, "... have different impacts on the aerosol distribution in China". Insert a suitable reference at this point.**

This statement is our inference based on the different effects of El Niño events with different durations on the meteorological fields over China and is what we showed as the main results and one of the highlights in this study. It has not been examined in any previous study.

**4. Why not used the observed $PM_{2.5}$ data during 2014-2017? Please check whether the number of stations is 1657.**

Since all other external forcings, including insolation, greenhouse gas concentrations, and emissions of aerosol and precursor are kept at year 2014 levels in E3SM modeling, we used the PM2.5 data from December 2014 to February 2015 (the nearest winter available) to evaluate the model performance in simulating winter aerosol concentration and distribution in China. We have confirmed the number of available stations is indeed 1657 over December 2014–February 2015.

**5. Please identify the simulation period in this study.**

As we described in the manuscript, the simulations in our study were driven by climatological average of monthly SST and SIC over 1870-2017 or monthly SST representing composite SD and LD El Niño events. All simulations were integrated for 20 years. All other external forcings, including insolation, greenhouse gas concentrations, and emissions of aerosol and precursor are kept at present-day conditions (year 2014). The simulations uing E3SM earth system model here are performed at present-day conditions instead of a certain year in chemical transport models or air quality models.

**6. It is recommended to list a table to introduce the experiments.**

Thanks for the suggestion. We have added the following table in the revised manuscript.

| Table 1. Experimental design. |                                                                       |
|-------------------------------|-----------------------------------------------------------------------|
| Experiments                   | Model Configuration                                                   |
| CLIM                          | Climatological SST                                                    |
| SD                            | Climatological SST + ΔSST SD El Niño                       |
| LD                            | Climatological SST + ΔSSTLD EI Niño                                   |
| SD_emis                       | Same as SD but turn off the emissions from South and Southeast Asia   |
| CLIM_emis                     | Same as CLIM but turn off the emissions from South and Southeast Asia |

7. Precipitation can exert notable scavenging effects on  $PM_5$  concentrations, whilst weak precipitation might increase  $PM_{2.5}$  concentrations by hygroscopic increase associated with increased humidity. Therefore, it is recommended to examine the relative humidity anomalies in section 3.2, which might be able to explain opposite pattern of precipitation and wet deposition anomalies.

It is possible that increase in relative humidity accelerates the chemical transformation of secondary aerosols (Yang et al., 2015), leading to the PM2.5 anomaly pattern and the positive relationship between the precipitation and aerosol concentration. Therefore, we showed the differences in DJF mean surface BC and POM concentrations in Fig. A. The spatial differences in primary aerosols of BC and POM between El Niño and climatological means and between SD and LD El Niño events are almost the same as PM2.5, indicating that change in water vapor is also not the main reason for the aerosol changes in China during El Niño events. We have added the description in the manuscript as "Water vapor can accelerate the chemical transformation of secondary aerosols (Yang et al., 2015), but the primary aerosols showed the same spatial differences in near-surface concentration as PM2.5 (not shown), indicating that change in water vapor is also not the main reason for the aerosol changes in China during El Niño events." In addition, the mass concentration showed in this study is dry aerosol mass. The water uptake will not influence the aerosol mass here.